# Perceived ability to comply with national COVID-19 mitigation strategies and their impact on household finances, food security, and mental well-being of medical and pharmacy students in Liberia

**Elvis J. Davis[1], Gustavo Amorim[2], Bernice Dahn[1], Troy D. Moon[3]***

1 University of Liberia College of Health Sciences, Monrovia, Liberia, 2 Department of Biostatistics, Vanderbilt University Medical Center, Nashville, Tennessee, United States of America, 3 Division of Infectious Diseases, Department of Pediatrics, Vanderbilt University Medical Center, Nashville, Tennessee, United States of America

* troy.moon@vumc.org

**Data Availability Statement:** All relevant data are within the paper and its Supporting Information files.

## Abstract

### Introduction

From the outset of the COVID-19 pandemic, guidance from WHO has promoted social distancing, wearing face masks, frequent hand washing, and staying-at-home as measures to prevent the spread of COVID-19. For many across Africa, compliance can be difficult. The aim of this study was to 1) understand the impact of student's household's ability to comply with COVID-19 mitigation strategies, 2) identify predictors of mitigation strategy compliance, and 3) describe the impact of COVID-19 on household economics, food-security, and mental well-being.

### Materials and methods

We conducted an email-based survey among current medical and pharmacy students of the University of Liberia College of Health Sciences between July and October 2020. The questionnaire was designed to explore their household's ability to comply with current mitigation strategies, as well as the pandemic´s impact on the student's household's finances and food security. Descriptive statistics were used to delineate demographic characteristics. Logistic regression was used to model factors associated with ability to comply with COVID-19 mitigation strategies, as well as participant's food security.

### Results

113 persons responded to the questionnaire. Seventy-six (67·3%) reported income losses as a result of the pandemic, with 93 (82·3%) reporting being "somewhat" or "very worried" about their households' finances. Seventy-seven (68·1%) participants reported food stocks that were sufficient for one-week or less. Forty (35%) participants reported eating less

**Funding:** This research was supported through the National Academies of Sciences, USAID PEER Liberia project (grant # AID-OAA-A-11-00012). Any opinions, findings, conclusions, or recommendations expressed are those of the authors alone, and do not necessarily reflect the views of USAID or NAS. The funders had no role in study design, data collection and analysis, decision to publish, or preparation of the manuscript.

**Competing interests:** The authors have declared that no competing interests exist.

preferred foods or skipping meals in the past week. Overall, 20 participants (19·4%) had a positive depression screen.

## Conclusions

Study participants showed mixed results in being able to adhere to national COVID-19 mitigation strategies, with household level stressors experienced around finances and food security. Until Liberia has access to vaccinations for most of its citizens, COVID-19 response measures need to provide social protections that address basic needs (shelter, clothing and food), and which specifically targets food insecurity. Preventative interventions for mental health problems must be incorporated into Liberia's response to the pandemic.

## Introduction

Severe acute respiratory syndrome coronavirus 2 (SARS-CoV-2) infection and associated novel coronavirus disease (COVID-19) remain an increasing global threat. As of March 6, 2021, approximately 116 million confirmed cases were reported by the World Health Organization (WHO) with 2.5 million deaths globally (case fatality rate [CFR] 2.2%). The Africa Region recorded 2·8 million cases and over 73,000 deaths (CFR 2.5%) during the same time period [1]. As COVID-19 cases began to spread geographically and enter Africa, initial predictions of its impact were dire due to weak health systems, limited human resources, and limited intensive care and mechanical ventilation capacity [2,3]. However, COVID-19's impact on Africa has remained puzzling, with fewer cases and lower CFR early on than originally predicted [4]. Several factors have been hypothesized to impact the trajectory and severity of COVID-19 in Africa including limited testing and laboratory infrastructure, a younger population, preexisting immunity, genetic factors, or possibly earlier implementation of preventive measures [4].

To fully understand the context of Liberia's response to the SARS-COV-2 pandemic, one must recognize that Liberia is facing its second major infectious disease outbreak in less than a decade, the first being the West African Ebola virus outbreak of 2014–2016. At the time the Ebola outbreak began, the capacity of Liberia's health system was severely limited. Essential functions such as the numbers of qualified health workers, infrastructure, logistics, health information, surveillance, and governance did not perform well, thus impeding a suitable and timely response to the outbreak [5]. As Liberia struggled with providing both emergency and routine care, the challenges in managing the outbreak were compounded by the deaths of front-line health workers and a declining morale, as well as growing distrust by affected populations in the system's ability to cope and respond accordingly [6].

From the outset of the COVID-19 pandemic, guidance from the WHO has promoted social distancing, wearing of face masks, frequent hand washing, and staying at home to prevent the spread of COVID-19. However, for many across Africa, compliance with these recommendations can be difficult [7]. Urban areas are at particularly high risk of COVID-19 transmission. They are frequently densely populated, with small informal dwellings, comprised of multi-generational households with shared sanitation facilities, a high level of social mixing, and transient residents [8,9]. Fragile health systems likely exacerbate the impact of the outbreak due to limitations in the ability to conduct adequate surveillance and control in low- and middle-income countries (LMIC) [10]. A lack of publicly available information and/or the spread of

misinformation further compound the situation by creating confusion and possible distrust of mitigation efforts [11].

On March 21, 2020, the government of Liberia declared a national state of emergency with mandatory school closings and lock downs of certain high-risk regions, including the capital, Monrovia; followed by national stay-at-home orders being issued as of April 10, 2020 [12]. As of December 2020, Ministry of Health (MOH) guidelines remained in place for all of Liberia's 15 counties, including the wearing of face masks in public places, promotion of social distancing of approximately 2 meters (6 feet), mandated hand washing stations at all operating businesses and services, closure of establishments serving alcohol by 9pm, and limitations at religious services to 25% capacity [13].

For this study, we conducted a cross-sectional email-based survey among currently enrolled medical and pharmacy students at the University of Liberia College of Health Sciences (ULCHS) in Monrovia. The aims of this survey were to 1) better understand the impact of COVID-19 on the student's household's ability to comply with COVID-19 mitigation strategies in place, 2) identify potential predictors of mitigation strategy compliance, and 3) describe the impact of COVID-19 on household economics, food-security, and mental well-being. With these results we hope to more specifically elucidate the country specific impacts of COVID-19 mitigation strategies on every day Liberians. These results will hopefully aid Liberia's policy makers in placing resources where they can provide the most help and provide guidance to other low-and middle-income country stakeholders that may be grappling with similar issues.

## Materials and methods

### Study design and participant recruitment

We conducted a cross-sectional email-based survey among current medical and pharmacy students of ULCHS between July 1, 2020 and October 31, 2020, through purposive sampling, based on email list-serves generated by the University. At the time of study enrollment, classes had been suspended and the majority of students had returned to their family homes, across Liberia, to ride out the stay-at-home orders with their families. Inclusion criteria included those students ≥ 18 years of age and those with active email accounts. We chose to utilize students during this unique time when they were home with their families as they represented a study population for which we had email addresses; were a population with a high likelihood of responding to an email survey request; and represented a mechanism through which it was felt we could get a quick snapshot of conditions in their family households to which they had returned.

An electronic questionnaire was created using the Research Electronic Data Capture (RED-Cap) platform. The questionnaire consisted of 66 closed-ended questions that were designed to elicit responses related to the participant's household's ability to comply with current infection control measures, as well as the pandemic´s impact on the household's finances and food security. Response options consisted of multiple-choice answers, true-false, and yes-no responses; as well as choices on a Likert scale such as "never", "sometimes", "often", and "always." Additional questions designed to screen the participant for depression were included using the eight question Personal Health Questionnaire Depression Scale (PHQ-8) with a focus on feelings and behaviors two weeks prior to being surveyed [14].

To assess food security, we asked participants two questions, one related to the current state of food stocks at their household and a second related to the participants food consumption in the 7 days prior to being surveyed. Response options included "I have had no difficulties eating enough food (normal pattern)"; "I ate less preferred foods"; I skipped meals or ate less than

usual"; "I have gone at least one full day without eating"; and "I have increased my food intake". We then classified participants as Food Insecure if they responded affirmatively to any of the responses with decreased food intake and Food Secure if they responded no changes to their normal eating pattern or increasing their food intake.

## Statistical analysis

Survey results were exported from REDCap and analyzed with the statistical software R (version 3.6.3; www.r-project.org). Descriptive statistics and group-wise comparisons were used to delineate demographic characteristics of respondents. Group-wise comparisons included Pearson's chi-squared test for categorical variables and nonparametric Mann-Whitney-Wilcoxon tests were used for continuous variables. Logistic regression was used to model factors associated with ability of households to comply with different COVID-19 mitigation strategies, as well as participant's food security.

Due to our small sample size, adjusting for all covariates may lead to overfit and unreliable inference. To restrict the number of parameters to be estimated, principal component analysis (PCA) for mixed (quantitative and qualitive) data were used, with varimax-rotation [15,16]. The first three components were used to summarize all variables and used as covariates in the logistic regression. With three covariates we guarantee at least 10 events per variable, providing reliable inference for the parameters of interest. The COVID-19 pandemic is a highly dynamic situation with rapid changes happening at a societal level and constantly changing measures, such that it was felt a study which could be performed quickly may still be useful to analyze the impact of those measures at that specific moment of the pandemic. While a larger sample would be preferable, it would demand more time such that the situational context of mitigation strategies and their impacts could be completely different at survey onset compared to survey end, complicating data analysis and interpretation of the results. This study, thus, provides a snapshot of COVID-19 and its implications in this specific group during a time where stay-at-home measures were enacted.

All variables used in the statistical analysis were selected *a priori*, based on our theoretical expectations. We did not use automatic selection procedures and variables were not selected based on p-values observed in univariable regression, as these could lead to unreliable inference. A significance level for all testing was two-sided and set at 0.05.

For the depression screening domain of the questionnaire, a score that ranged from 0–24 was calculated, with each question generating a score of 0–3. When the total score was $\geq 10$, the participant was screened as positive for depression [14,17]. A semiparametric ordinal regression model was used to assess factors associated with higher depression scores. All estimates were presented in terms of point estimates and 95% confidence intervals (CI).

## Ethical considerations

The survey's email invitation included language inviting the recipient to participate in the study. If they chose to advance, the email then took them to an electronic informed consent form that they were asked to read and sign before advancing to the survey itself. The study protocol was approved by the Institutional Review Boards of the University of Liberia-Pacific Institute for Research and Evaluation (# 0-07-220) and Vanderbilt University Medical Center (#201005).

## Results

A study invitation and questionnaire were emailed to a list-serve of 265 currently enrolled pharmacy and medical students, of which 113 (43%) responded by signing the electronic

informed consent and completing the questionnaire. The median age was 28 years (interquartile range IQR = [26, 32]). Seventy (61.9%) respondents were men and 85 (75.2%) reported being single never married. The majority reported (34.5%) living in a single-unit property, with electricity (84.1%) and a place for handwashing (92%), with a median of seven [IQR: 5,10] persons living in the household (Tables 1 and S1).

**Table 1. Socio-demographic characteristics of survey respondents.**

| Characteristics | |
|---|---|
| **N = 113** | **N (%)** |
| Age Median [IQR| | 28 [26,32] |
| Gender | |
| Male | 70 (61.9) |
| Female | 43 (38.1) |
| Marital Status | |
| Single never married | 85 (75.2) |
| Married | 16 (14.2) |
| Cohabitating | 12 (10.6) |
| Occupation | |
| Employed for wages | 4 (3.5) |
| Self employed | 2 (1.8) |
| Unemployed | 1 (0.9) |
| Student | 105 (92.9) |
| Other | 1 (0.9) |
| Highest Education Completed | |
| Highschool graduate | 2 (1.8) |
| College 1–3 years | 6 (5.3) |
| College 4 years or more | 105 (92.9) |
| Number of people living in the household | 7 [5,10] |
| Including yourself, number of persons > 60 years old in household Median [IQR] | 0 [0,1] |
| Including yourself, number of persons ≤ 5 years old in household Median [IQR] | 1 [0,2] |
| Are you currently pregnant | |
| Yes | - |
| No | 42 (37.2) |
| Not applicable/Missing | 71 (62.8) |
| Ever been told by a health care professional you have had any of the following | |
| Heart attack | - |
| Angina or coronary artery disease | - |
| High blood pressure | 2 (1.8) |
| Type II diabetes | 1 (0.9) |
| Cancer | 1 (0.9) |
| Asthma | 4 (3.5) |
| COPD | - |
| Kidney disease | 1 (0.9) |
| HIV | - |
| Tuberculosis | 1 (0.9) |
| Ebola Virus disease | - |
| Lassa Fever | - |
| Immunodeficiency | - |
| None of above | 104 (92.0) |

## Compliance with COVID-19 mitigation strategies

A total of 89 (78.7%) participants reported being either "very worried" or "somewhat worried" about the health of their household members because of the COVID-19 pandemic, yet 77 (68.1%) reported they were not able to follow stay-at home recommendations. In the two-weeks prior to being surveyed, 59 (52.2%) participants reported the need to leave their house to purchase goods between "1–2 times per week" and "at least once per day", with 50 (44.2%) participants reporting they felt they could "never" or only "sometimes" practice social distancing on these outings. In contrast, 91 (80.5%) participants reported face mask usage either "often" or "always" when outside of their homes (S2 Table).

## Household economics and food security

We asked participants about their household economics in the month prior to being surveyed (S3 Table). Seventy-six (67.3%) reported they had experienced income losses as a result of the pandemic, with 93 (82.3%) participants reporting they were either "somewhat" or "very worried" about their households' financial situation. Only 15 (13.3%) participants felt they could maintain Liberia's stay-at-home recommendations for as long as was needed without being financially impacted.

Food security was another issue of worry, with 77 (68.1%) participants reporting insufficient food stocks at their household or provisions that were only sufficient to last for one-week or less. Additionally, 40 (35%) participants reported that in the week prior to being surveyed they had altered their daily food consumption, eating less preferred foods or skipping meals all together.

We conducted univariate and multivariable logistic regression to explore the impact of different variables on the ability of households to comply with COVID-19 mitigation strategies, as well as their food security. Univariate comparisons suggested that older participants were more likely (p = 0.02) to adhere to social distancing recommendations "often" or "every time" they left their household and more likely (p = 0.03) to comply with stay-at-home recommendations. Participants that lived with a partner (married or cohabitating), trended toward being more likely (p = 0.07) to wear face masks when out in public, but showed no association with either social distancing or compliance with stay-at-home recommendations. Further, sharing a household with more people was somewhat associated (p = 0.09) with higher compliance with stay-at-home recommendations. Participants that reported being "very worried" about their household's health trended towards better social distancing adherence when outside the home (p = 0.06) as well as a higher compliance with stay-at-home recommendations (p = 0.10) (Table 2). Finally, living in households with more people had a higher likelihood (p = 0.06) of the participant being food insecure. Men were less likely (p = 0.05) to report experiencing food insecurity (Table 3).

A multivariable logistic regression was used to explore factors associated with adherence to COVID-19 mitigation recommendations and food security (Table 4). In order to not overfit the model, we first ran a principal component analysis (PCA) with mixed data to reduce the number of parameters into combinations with the best possible correlation, followed by a multivariable logistic regression using the first three principal components (PC1, PC2, PC3) as covariates. The loadings corresponding to PC1, PC2, and PC3 that are used in the analysis of compliance to face masks, social distancing, and to stay-at-home measures, are presented in S4 Table. The first principal component, PC1, was related to participants that were, on average, older and living with a partner; PC2 was related to participants that were sharing the house with several other people and were, on average, "very concerned" about the health of their household; while PC3 was related to men. Similar steps were taken to find risk factors

**Table 2. Univariate associations with wearing face masks in public, practicing social distance, or compliance with stay-at-home recommendations.**

| | Face Mask | | | Social Distance | | | Stay-at-Home* | | |
|---|---|---|---|---|---|---|---|---|---|
| | Never Sometimes Often | Every time | p- | Never Sometimes | Often Every time | p- | Poor | Good | p- |
| Age | 28 [26,32] | 29 [26,32] | 0.89 | 28 [25,30] | 30 [27,32] | 0.02 | 28 [26,30] | 30 [27,32] | 0.03 |
| Male | 22 (55) | 44 (65) | 0.37 | 26 (53) | 40 (69) | 0.14 | 28 (54) | 38 (69) | 0.16 |
| Married/cohabitating | 6 (15) | 22 (33) | 0.07 | 11 (22) | 17 (29) | 0.56 | 13 (25) | 15 (27) | 0.96 |
| Number living in household | 8 [4,10] | 7 [5,10] | 0.87 | 7 [5,9] | 8[5,12] | 0.18 | 6.5 [5,9] | 8 [6,12] | 0.09 |
| Health: Very Worried | 22 (55) | 29 (43) | 0.33 | 18 (37) | 33 (57) | 0.06 | 20 (39) | 31 (56) | 0.10 |
| Loss of income: Yes | 26 (65) | 49 (73) | 0.50 | 32 (65) | 43 (74) | 0.43 | 34 (65) | 41 (75) | 0.41 |

*Stay-at-Home = compliance with stay-at-home recommendations.

Reference levels: Gender: Female; Marital status: Single/living alone; Health: Not worried; Loss of income: No. Categorical variables are presented in frequencies (%) and continuous variables in median and interquartile range. P-values were computed with Chi-square tests for categorical variables and the Mann-Whitney-Wilcoxon rank test for continuous variables.

associated with food security; a principal component regression analysis was performed, and the first three components were used as covariates. Their loadings are presented in S5 Table, suggesting that PC1 was again related to participants that were, on average, older and living with a partner; PC2 was related to participants that were sharing the house with several other people; and PC3 was related to participants with electricity in their home. In multivariable logistic regression, we found participants that reported living in a household with a larger number of people AND that reported being "very worried" about the health of their household were more likely to practice social distancing when they left their household (OR: 1.48; 95% CI: 1.03–2.23; p = 0.04) and more likely to comply with stay-at-home recommendations (OR: 1.50; 95% CI: 1.05–2.24; p = 0.03). Men, on the other hand, were 27% less likely to practice social distancing or comply with stay-at-home recommendations, though this did not quite reach statistical significance (OR:0.73; 95% CI: 0.50–1.04; p = 0.08). Participants that were older and living with a partner (OR: 1.44; 95% CI: 1.04–2.03; p = 0.03); and those that shared their household with more people (OR: 1.81; 95% CI: 1.20–3.00; p = 0.01), were both significantly more likely to be food insecure.

## Depression

Overall, 103 participants responded to the questions making up the PHQ-8 depression screening. The majority were men (61%), single (75%), with a median age of 29 years [IQR: 26, 32], and a median reported number of 7 persons living in their households [IQR:5,10]. Sixty percent reported being "very worried" about their finances, 72% were "very worried" about losses

**Table 3. Univariate associations with participant food security.**

| | Food Secure | Food Insecure | p-value |
|---|---|---|---|
| Age | 28 [26,31] | 29 [27,33] | 0.07 |
| Male | 36 (54) | 30 (75) | 0.05 |
| Number living in household | 7 [4,10] | 8 [6,13] | 0.06 |
| Electricity: Yes | 59 (88) | 32 (80) | 0.40 |
| Loss of income: Yes | 42 (63) | 33 (83) | 0.05 |

Reference levels: Gender: Female; Electricity: No; Loss of income: No. Categorical variables are presented in frequencies (%) and continuous variables in median and interquartile range. P-values were computed with Chi-square tests for categorical variables and the Mann-Whitney-Wilcoxon rank test for continuous variables.

**Table 4. Multivariable logistic regression: Factors associated with adherence to COVID-19 mitigation recommendations and food security.**

| | Face Mask | | | Social Distance | | | Stay-at-Home* | | | Food Security | | |
|---|---|---|---|---|---|---|---|---|---|---|---|---|
| | OR | 95% CI | p-value | OR | 95% CI | p-value | OR | 95% CI | p-value | OR | 95% CI | p-value |
| PC1 | 1.26 | 0.90–1.79 | 0.18 | 1.31 | 0.94–1.85 | 0.12 | 1.22 | 0.88–1.71 | 0.24 | 1.44 | 1.04–2.03 | 0.03 |
| PC2 | 0.86 | 0.60–1.21 | 0.38 | 1.48 | 1.03–2.23 | 0.04 | 1.50 | 1.05–2.24 | 0.03 | 1.81 | 1.20–3.00 | 0.01 |
| PC3 | 0.84 | 0.58–1.20 | 0.33 | 0.73 | 0.50–1.04 | 0.08 | 0.73 | 0.50–1.04 | 0.08 | 1.06 | 0.70–1.59 | 0.77 |

*Stay-at-Home = Compliance with stay-at-home recommendations.

For Face Mask, Social Distancing, and Compliance: PC1: Participants that were, on average, older and living with a partner; PC2: Participants that were sharing the house with several other people and were, on average, "very concerned" about the health of their household; PC3: Men.

For Food Security: PC1: Participants that were, on average, older and living with a partner; PC2: Participants that were sharing the house with several other people; PC3: Participants reporting electricity in their home.

in income, and 48% were "very worried" about the health of their households as a result of COVID-19. Overall, 20 participants (19.4%) had a positive depression screen with a PHQ-8 score $\geq$ 10. Univariate comparisons suggest that concerns about the health of household members, household finances, and sharing a house with more people were associated with a higher odds of screening positive for depression (S6 Table). A multivariable analysis could not be performed due to sample size.

Next, we ran univariate and multivariable ordinal regression analysis (without dichotomizing depression score) to assess the impact of *a priori* selected variables on depression score. Variables included gender, marital status, number of people in household, as well as concern over household health, loss of income, and finances. S1 Fig shows how these variables correlate to depression score. Compared to women, men were, on average, more likely to have higher depression scores, although no statistical significance was found. Similar findings were seen for participants living with a partner and those worried about their finances. In multivariable ordinal regression (Table 5), those factors associated with a positive PHQ-8 depression screen were being "very worried" about the health of one's household (OR: 2.43; 95% CI: 1.08–5.46; p = 0.03) and about one's household finances (OR: 2.27; 95% CI: 0.96–5.37; p = 0.06).

## Discussion

Liberia declared a state of emergency and implemented national stay-at-home orders on April 10, 2020, that remained in effect through the end of December 2020. In our cohort, only 13% of respondents reported that they felt they could fully adhere to the stay-at-home orders for as

**Table 5. Logistic regression: Factors associated with a positive PHQ-8 depression screen.**

| | Univariate Regression | | | Multivariable Regression | | |
|---|---|---|---|---|---|---|
| | OR | 95% CI | p-value | OR | 95% CI | p-value |
| Age | 1.03 | 0.97–1.11 | 0.34 | 0.95 | 0.87–1.04 | 0.27 |
| Male | 1.71 | 0.84–3.50 | 0.14 | 1.70 | 0.78–3.70 | 0.17 |
| Married/cohabitating | 1.41 | 0.65–3.02 | 0.38 | 1.86 | 0.78–4.43 | 0.16 |
| Number living in household | 1.04 | 1.01–1.08 | 0.02 | 1.03 | 0.99–1.07 | 0.17 |
| Health: Very Worried | 3.47 | 1.70–7.08 | <0.001 | 2.43 | 1.08–5.46 | 0.03 |
| Loss of income: Yes | 1.35 | 0.62–2.94 | 0.45 | 1.14 | 0.49–2.64 | 0.76 |
| Finances: Very Worried | 3.38 | 1.63–7.00 | <0.001 | 2.27 | 0.96–5.37 | 0.06 |

OR = odds ratio; CI = confidence interval.

Reference levels. Gender: Female; Marital status: Single/living alone; Health concerns: Not worried; Loss of income: No; Finances: Not worried.

long as was needed. In fact, about half of participants reported they currently leave the house multiple times during the week to purchase goods and/or to go to work. When asked about the ability to social distance when out of the home, only about 50% reported that they felt they could do this often or every time. However, in contrast, face mask uptake was quite high, with >80% reporting use of a face mask often or every time they leave the home.

Since the beginning of the SARS-CoV-2 pandemic, countries around the world have struggled with balancing the positive public health gains from mitigation strategies, against the negative economic and social costs these strategies can produce [18,19]. As rich countries begin to see a light at the end of the tunnel with larger proportions of their populations being vaccinated; low resource countries across Africa, South America, and Asia may have to wait until 2023 before widespread immunization reaches a level in which mitigation strategies can be safely rolled back [20]. As such, greater understanding of the effects of mitigation strategies on a given population are needed so that appropriate stop-gap measures can be put in place to support these vulnerable populations until vaccination roll-out can be fully realized. To the best of our knowledge, this research is one of the first studies describing the ability of Liberian households to comply with national mitigation strategies and the impact the pandemic is having on their finances, food security, and individual well-being.

Liberia is one of the world's poorest countries as a result of civil conflict, major infectious disease emergencies, and overall poor governance [21,22]. According to the United Nations Development Program (UNDP), Liberia ranked 175 of 189 countries on the 2020 Human Development Index (HDI) [23]. Following the end of Liberia's civil wars in 2003, the country began to show steady progress in terms of economics, health, and other key development indicators [24]. However, this progress quickly stalled due to the 2014–2016 West African Ebola outbreak [22]. During the 2-year period of active Ebola spread, Liberia lost ~40% of wage-earning jobs [25]. In the years since, Liberia once again is rebuilding, yet as of the end of 2019, it was estimated that roughly 16% of Liberia's population was food insecure and 83% lived in extreme poverty [26]. In order to understand how household economics and food security may have been impacted further as a result of the COVID-19 pandemic, we questioned participants about their household finances and found that nearly two-thirds were reporting losses in income as a result of the pandemic. Furthermore, this generated considerable stress, with approximately 56% of respondents reporting they were "very worried" about their household's finances. We also questioned participants about their household's food security based on the quantity of household food stocks and how long they would last; as well as a question about whether the participant's personal food consumption had changed in the week prior. We found that nearly two-thirds of households had food stocks that were only sufficient for one-week or less, and roughly one-third of participants reported decreases in their personal food consumption from what they considered normal. The risk to food security across Africa as a result of COVID-19 has been well described. Many African countries are net importers of food for consumption, with their own agricultural production being prioritized for commercial exportation. During situations of emergency, this dynamic can result in both shortages to the local food supply as well as skyrocketing prices [18,27]. Countries such as Liberia are especially vulnerable, and our findings highlight that COVID-19 mitigation strategies are likely contributing to worsening household level food security, or on the contrary, that fears about food insecurity are forcing households to make decisions that increase their risk of contracting SARS-CoV-2, due to inability to fully adhere to COVID mitigation strategies. This fact becomes more worrisome as Liberia has been identified as 1 of 10 countries in which a longer-term state of national undernutrition may be a significant driver of high COVID-19 mortality rates in the country [28]. The Government of Liberia, and its partners such as the World Food Program, have begun to address food security problems through the COVID-19 Household

Food Support Program (COHFSP), targeting nearly 50,000 households for provision of staple food commodities as well as programs specifically targeting rural women and school aged children [29]. While an important first step, it's likely that much greater investments toward food security and other social protections will urgently be required throughout Liberia's COVID-19 response and in the years following.

The COVID-19 pandemic has generated well documented mental and psychological health problems around the world [30,31]. Liberia is no different, with slightly more than 19% of study participants screening positive for depression. Our study identified being "very worried" about household member's health as well as being "very worried" about household finances, to be highly correlated with a positive depression screen. These correlates make sense and are consistent with other studies which found that COVID's perceived or real impact on one's control over their daily life, predicted negative psychologic consequences [32,33]. Over the last 30 years, Liberia has suffered multiple traumatic events. A study in 2008, five years after the end of Liberia's civil wars, found that 44% of Liberians suffered major depression and 40% had post-traumatic stress disorder (PTSD) [34]. Ten years later, roughly 20% of Liberia's Ebola survivors were reported to screen positive for depression and another 10% screened positive for general anxiety disorder [35]. Mental health problems were a problem in Liberia even before the COVID-19 pandemic. Providing mental health services has been challenging, as national mental health expenditures average about US$0.02 per person and there is currently < 1 psychiatrist and <1 mental health nurse per 100,000 population in Liberia [35].

This study has several limitations. First, we tried to address safety concerns during the pandemic by conducting our survey by means of an email list-serv. This resulted in only 43% of potential participants completing the survey and thus limits the generalizability of study findings. We tried to select a population that was easily reachable through email and which could also provide household level information, as many respondents were expected to be back at their family homes at the time of the survey. However, by targeting students, our data are skewed towards younger respondents [IQR: 26,32 years]. Further, by targeting only households in which a medical or pharmacy student resides, we are likely limiting the generalizability of our results to households that may be more socio-economically advantaged compared to the general population. Next, we tried to explore and highlight household food security in our population based on two simple questions related to the quantity of existing food stocks as well as changes in one's food consumption. Many examples exist of more nuanced strategies and questioning for determining food security at individual, household, and country levels. Our questioning provides only a glimpse into this issue for our respondents and should be followed-up with more in-depth study. Finally, as this was a cross-sectional survey, no causal inference can be made as to the associations we highlight.

## Conclusion

Study participants showed mixed results in terms of adherence to national COVID-19 mitigation strategies. Many have doubts as to the length of time they can maintain stay-at-home orders and reported limited ability to practice social distancing when out of the home. Despite this, Liberians show a willingness to comply when it is feasible, as highlighted by >80% face mask usage when out of the home. COVID-19 is putting stress on household finances and more than a third of respondents reported eating less preferred foods or skipping meals in the week prior to being surveyed. Positive depression screening was common and associated with intense worry about household member health and household finances. Until such time as Liberia has access to vaccinations for a majority of its citizens, national COVID-19 response measures need to provide social protections that address basic needs (shelter, clothing and

food), and which specifically targets household level food security and ensuring maintenance of good nutrition. Preventative interventions for mental health problems must be incorporated into Liberia's response to the pandemic.

## Supporting information

**S1 Fig. Associations between each covariate with depression score.**
(TIF)

**S1 Table. Characteristics of household and additional persons living in household.**
(DOCX)

**S2 Table. Compliance with COVID-19 mitigation recommendations.**
(DOCX)

**S3 Table. Financial and food security concerns resulting from the COVID-19 pandemic.**
(DOCX)

**S4 Table. Loadings from principal component analysis with mixed data, combining the following variables: Age, number of people in house, marital status, health concerns, loss of income, and gender.** C1: First component from the principal component analysis with mixed data; C2: Second component; C3: Third component.
(DOCX)

**S5 Table. Loadings from principal component analysis with mixed data, combining the following variables: Age, number of people in house, marital status, loss of income, and electricity.** C1: First component from the principal component analysis with mixed data; C2: Second component; C3: Third component.
(DOCX)

**S6 Table. Factors associated with a positive depression screen (PHQ-8 $\geq$10).** Categorical variables presented as absolute value (%); continuous variables presented via medians and interquartile range. Reference values: Gender, female; marital status, single/never married; worried about health, not worried/somewhat worried; loss of income, no/not applicable; worried about finances, not worried/somewhat worried; p-value computed via chi-square tests for categorical variables and Mann-Whitney-Wilcoxon test for numerical variables.
(DOCX)

**S1 File. Study questionnaire.**
(PDF)

## Acknowledgments

We would like to thank the student body of the University of Liberia College of Health Sciences for their participation in this study.

## Author Contributions

**Conceptualization:** Elvis J. Davis, Bernice Dahn, Troy D. Moon.

**Data curation:** Elvis J. Davis.

**Formal analysis:** Elvis J. Davis, Gustavo Amorim.

**Funding acquisition:** Troy D. Moon.

**Investigation:** Troy D. Moon.

**Methodology:** Gustavo Amorim, Troy D. Moon.

**Supervision:** Bernice Dahn.

**Writing – original draft:** Elvis J. Davis, Troy D. Moon.

**Writing – review & editing:** Gustavo Amorim, Bernice Dahn, Troy D. Moon.

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
