## [Decision Letter · Decision Letter 0]

19 May 2021

PONE-D-21-08122

Perceived ability to comply with national COVID-19 mitigation strategies in Liberia and their impact on household finances, food security, and mental well-being

PLOS ONE

Dear Dr. Troy,

Thank you for submitting your manuscript to PLOS ONE. After careful consideration, we feel that it has merit but does not fully meet PLOS ONE’s publication criteria as it currently stands. Therefore, we invite you to submit a revised version of the manuscript that addresses the points raised during the review process.

We look forward to receiving your revised manuscript.

Kind regards,

Shah Md Atiqul Haq

Academic Editor

PLOS ONE

Additional Editor Comments:

Dear Author,

I would like to ask you to revise the paper. If you agree to revise and resubmit, then I will resend the paper to the reviewers again.

Good luck!

Journal Requirements:

2. Please include additional information regarding the survey or questionnaire used in the study and ensure that you have provided sufficient details that others could replicate the analyses. For instance, if you developed a questionnaire as part of this study and it is not under a copyright more restrictive than CC-BY, please include a copy, in both the original language and English, as Supporting Information.  If the original language is written in non-Latin characters, for example Amharic, Chinese, or Korean, please use a file format that ensures these characters are visible.

Reviewers' comments:

Reviewer's Responses to Questions

**Comments to the Author**

1. Is the manuscript technically sound, and do the data support the conclusions?

Reviewer #1: Yes

Reviewer #2: No

2. Has the statistical analysis been performed appropriately and rigorously? 

Reviewer #1: Yes

Reviewer #2: I Don't Know

3. Have the authors made all data underlying the findings in their manuscript fully available?

Reviewer #1: Yes

Reviewer #2: Yes

4. Is the manuscript presented in an intelligible fashion and written in standard English?

Reviewer #1: Yes

Reviewer #2: Yes

5. Review Comments to the Author

Reviewer #1: At first, when I read the text, I was surprised that it was about surveying students, because that was not mentioned in the abstract. The authors should better present that already there. Because this is crucial for the interpretation of the results. The fact that, for example, compliance with mask wearing is high could be explained by educational inequality.

The study aims to describe the ability of Liberian households to comply with national mitigation strategies and the impact the pandemic is having on their finances, food security, and individual well-being. Although the limitations of the study state that students are not representative of the population as a whole due to their average age, they do not state that students and their family housholds per se represent a bias, since not everyone has access to higher education. I know very little about the education system in Liberia, unfortunately, but I suspect that access to university is highly socially selective for children from wealthier families. The authors should elaborate a bit on this.

The response rate is high, but couldn't a non-responder analysis have provided a bit more coverage? The student data are known to the university administration and are certainly available in anonymized form. It would have been possible to show whether the realized sample is selectively biased, e.g. with regard to age or gender distribution.

Because of the things that have been mentioned, the study does report very relevant and important results, but I think one should not overestimate the scope of the study. For one thing, it just (the case of Ebola) suggests why the case of Liberia is a good starting point to make statements about the impact of the COVID pandemic in African countries. But in the introduction, I think there is too much focus on continents that are structurally disadvantaged in the pandemic. Here, it might be better to elaborate a bit more clearly on what can be learned from this country study. It should also be refelected that the acquisition via email and the resulting selection of the study population entails major limitations in the distribution range. The statistics here suggest too great a degree of objectively generalizable facts.

Reviewer #2: This paper addresses the very important, original, and timely question of what drives households’ ability to comply with COVID restrictions and the latter’s impact on food insecurity, mental health, and finances in Liberia. The authors address this question using data collected from a survey of medical/ pharmacy students in Liberia, and by calculating descriptive statistics and conducting logistic regressions. The authors find that face mask use is high (>80%) but the ability to comply with social distancing was relatively low and only 13% reported that they felt they could fully adhere to the stay-at-home orders. Additional analyses explore socio-demographic correlates of these outcomes as well as poor mental health and food insecurity.

The study presents the results of original research, the results of which do not appear to have been reported elsewhere. The article is generally presented in an intelligible fashion and is written in standard English, although there are some important areas requiring clarification. However, I have some concerns about the paper. The sample size is extremely small and the authors need to do more to discuss the generalizability of the data. Furthermore, some of the analyses (methods and results) are difficult to understand and the conclusions could be foreshadowed with greater discussion of the literature motivating the variables under study.

Specific comments:

1. Abstract- ‘introduction’ should introduce the goals/ objectives of the paper.

2. Title and abstract should reflect the focus on students in the data – e.g. substitute ‘household’ for ‘students’ in both.

3. p.4-5 Should do much more to motivate the importance of the study population -why in particular would survey responses from this student population be of interest? I imagine the authors could develop an argument for studying this population (e.g. because of wider discussion about how youth are most adversely affected by the pandemic, and/or importance of studying healthcare professionals), but this needs to be stated explicitly.

4. Related, the Introduction also needs to do more to justify the focus on the socio-demographic variables ultimately examined in the statistical analyses. Presumably the decision to focus on e.g. age, gender, and concern about household members’ health was motivated by some theoretical expectation in the literature. Please outline in the Introduction. This would greatly improve the reader’s interpretation of the results/ overall takeaways of the statistical analyses.

5. Food vulnerability and food insecurity are used interchangeably in the paper (e.g. on p.6). The authors should be consistent in the terminology used. I also encourage the authors to define the term where they use it – this is important given ambiguous meaning of this term and varying definitions used in the literature.

6. Coding of ‘food vulnerability’ needs more justification, and greater alignment with other studies. One common approach in the literature is to categorize scales such as that used in this paper into moderate and moderate/severe food insecurity, see e.g Barlow et al. 2020 (Reference also provides some discussion of the meaning of food insecurity and relevant references to additional literature that could be useful to cite here – see my point no.5 above).

Barlow, P., Loopstra, R., Tarasuk, V. and Reeves, A., 2020. Liberal trade policy and food insecurity across the income distribution: an observational analysis in 132 countries, 2014–17. The Lancet Global Health, 8(8), pp.e1090-e1097.

7. p.7 “Due to our sample size…” presumably the authors mean ‘due to our small sample size’? Please clarify. Please also provide more detail about the issues this raises and why PCA to identify three covariates was deemed appropriate. This will be helpful for readers less familiar with this approach and to provide a clearer justification for the statistical approach ultimately taken.

8. With respect to the small sample size, I’d like to see the authors do much more to demonstrate whether or not this is of concern. For example, can the authors compare some of the socio-demographics of respondents in their data with socio-demographics of the wider student population based on other larger scale surveys?

9. The small sample is the greatest weakness of the paper and whilst I appreciate that rapid surveys are also necessary to conduct timely analyses of important questions such as this, the authors need to do more in the Methods to explicitly address this issue. The authors could point to several advantages of a small sample – e.g. it permitted a timely analysis, it was the largest sample feasible within the resources provided, it can provide potentially useful exploratory insights, potentially other reasons?

10. With respect to the presentation of the results, this is generally ok. Table 4 is difficult to read and I would like to see some text in the left hand column explaining what C1-C3 are. This would greatly improve the readability of the table which is at present difficult to understand.

11. In the Discussion, please begin with a summary of the key results. This would greatly improve the discussion by demonstrating how many of the interesting points raised here are linked to the specific results and takeaways of this particular paper.

6. PLOS authors have the option to publish the peer review history of their article (what does this mean?). If published, this will include your full peer review and any attached files.

Reviewer #1: No

Reviewer #2: No

---

## [Author Response · Author response to Decision Letter 0]

14 Jun 2021

Please see the new uploaded Cover Letter with response to reviewers

---

## [Decision Letter · Decision Letter 1]

28 Jun 2021

Perceived ability to comply with national COVID-19 mitigation strategies and their impact on household finances, food security, and mental well-being of medical and pharmacy students in Liberia

PONE-D-21-08122R1

Dear Dr. Troy,

We’re pleased to inform you that your manuscript has been judged scientifically suitable for publication and will be formally accepted for publication once it meets all outstanding technical requirements.

Kind regards,

Shah Md Atiqul Haq

Academic Editor

PLOS ONE

Additional Editor Comments (optional):

Dear Authors,

The article has been accepted now.

Congratulations!!!

Reviewers' comments:

Reviewer's Responses to Questions

**Comments to the Author**

1. If the authors have adequately addressed your comments raised in a previous round of review and you feel that this manuscript is now acceptable for publication, you may indicate that here to bypass the “Comments to the Author” section, enter your conflict of interest statement in the “Confidential to Editor” section, and submit your "Accept" recommendation.

Reviewer #1: All comments have been addressed

2. Is the manuscript technically sound, and do the data support the conclusions?

Reviewer #1: Yes

3. Has the statistical analysis been performed appropriately and rigorously? 

Reviewer #1: Yes

4. Have the authors made all data underlying the findings in their manuscript fully available?

Reviewer #1: Yes

5. Is the manuscript presented in an intelligible fashion and written in standard English?

Reviewer #1: Yes

6. Review Comments to the Author

Reviewer #1: Thank you for responding appropriately to my comments. Even though the scope of the study is limited, I consider the publication important because the challenges posed by the pandemic in African countries should be given more scientific attention. Here, the article provides good and instructive hints not only for health policy, but also for further research.

7. PLOS authors have the option to publish the peer review history of their article (what does this mean?). If published, this will include your full peer review and any attached files.

Reviewer #1: No

---

## [Editor Report · Acceptance letter]

30 Jun 2021

PONE-D-21-08122R1 

Perceived ability to comply with national COVID-19 mitigation strategies and their impact on household finances, food security, and mental well-being of medical and pharmacy students in Liberia 

Dear Dr. Moon:

I'm pleased to inform you that your manuscript has been deemed suitable for publication in PLOS ONE. Congratulations! Your manuscript is now with our production department. 

Kind regards, 

on behalf of

Dr. Shah Md Atiqul Haq 

Academic Editor

PLOS ONE